# RT-IVT method allows multiplex real-time quantification of in vitro transcriptional mRNA production

Fengyu Zhang[1,4], Yipeng Wang[1,4], Xiaomeng Wang[1], Hongjie Dong [2], Min Chen[1], Ning Du[3], Hongwei Wang[1], Wei Hu[1], Kundi Zhang [1✉] & Lichuan Gu [1✉]

For the past 30 years, in vitro transcription (IVT) technology has been extensively used for RNA production or for basic transcriptional mechanism research. However, methods for mRNA quantification still need to be improved. In this study, we designed a RT-IVT method using binary fluorescence quencher (BFQ) probes and the PBCV-1 DNA ligase to quantify mRNA production in real-time by fluorescence resonance energy transfer (FRET) and RNA-splinted DNA ligation. Compared with existing methods, the RT-IVT method is inexpensive and non-radioactive, and can detect mRNA production in unpurified systems in real-time and shows high sensitivity and selectivity. The activity of T7 RNA polymerase and *Escherichia coli* RNA polymerase holoenzyme was then characterized with this method. We then multiplexed the real-time mRNA quantification for three T7 promoters on a RT-PCR thermocycler by using BFQ probes with different colored fluorophores that were specific for each target. Ultimately, we created an inexpensive multiplexed method to quantify mRNA production in real-time, and future research could use these methods to measure the affinity of transcriptional repressors to their target DNA sequence.

---

[1] State Key Laboratory of Microbial Technology, Shandong University, 72 Binhai Road, 266237 Qingdao, China. [2] Shandong Institute of Parasitic Diseases, Shandong First Medical University & Shandong Academy of Medical Sciences, 11 Taibaizhong Road, 272033 Jining, China. [3] Institute of Ecology and Biodiversity, School of Life Sciences, Shandong University, 72 Binhai Road, 266237 Qingdao, China. [4]These authors contributed equally: Fengyu Zhang, Yipeng Wang. ✉email: kdzhang@sdu.edu.cn; lcgu@sdu.edu.cn

In vitro transcription (IVT) is a classic procedure that allows for the synthesis of template-directed RNA molecules. Because RNA synthesis is simple, it has been used for the production of hybridization probes in RNase protection or interference experiments, the generation of antisense reagents during RNA-binding protein analysis, and the design of functional molecules for cell pluripotency research[1–3]. Additionally, IVT has also been extensively used to explore the basic mechanisms of transcription, remove excessive nuclear environments, allow for the study of gene promoter activity, and research gene-specific transcription regulators or RNA polymerase[4–6]. Since the outbreak of COVID-19, IVT-mRNA-based vaccines have become standard because of their safety profiles, as well as their precise and reproducible immune responses[7,8]. In addition to their safety and accuracy, IVT-mRNA procedures are also more cost-effective and faster than current RNA synthesis procedures, which could be multiplexed to combine their positive attributes.

With IVT, the system includes a DNA template, RNA polymerase, and nucleoside triphosphates (NTPs)[9]. During the reaction, the activity of the polymerase is determined by the quantification of mRNA product. However, mRNA strands have poor thermal stability and are rapidly degraded by ubiquitous RNases in the environment, which causes the product detection to be a limiting factor for efficient IVT applications.

In the case of high-yield transcription assays, such as the *Escherichia coli* phages T3 or T7 polymerase, mRNA production is detected by undergoing RNA purification and then quantified by agarose gel electrophoresis with SYBR Green staining, spectrophotometry, or reverse transcription and real-time quantitative polymerase chain reaction assays (RT-PCR)[10,11]. However, all these methods need to undergo RNA purification, which leads to the loss of total RNA yield, and is not suitable for small volume transcription assays. Another common IVT detection method uses radioactively tagged nucleotides (i.e., [$\alpha$-$^{32}$P] UTP mixed with ATP, CTP, and GTP), and denatured gel electrophoresis and autoradiography[12]. Ultimately, this method improved mRNA detection sensitivity and did not require purification, however, the radioactively tagged nucleotides are expensive and require appropriate safety measures, as well as a scintillation counter, which is restrictive to many labs. More importantly, stop reaction was needed before detection for each method, which makes real-time monitoring of transcription impossible. Therefore, an inexpensive, simple, universal and continuous detection approach is urgently needed.

For years, researchers have been trying to develop real-time detection for IVT assays, and the most recent breakthrough utilizes fluorescence resonance energy transfer (FRET) aptamers, such as molecular beacons (MBs) and binary probes[13–15]. MBs have been widely used in mRNA localization in vivo, but are rarely used in IVT unpurified systems because they create a false-positive signal from the interaction between their special loop-stem structure and the high concentration of nucleic acid binding proteins, like RNA polymerase and other transcription regulatory factors[16–20]. Fortunately, binary probes circumvent this defect and have been successfully used in real-time detection of IVT assays. Binary probes consist of two DNA oligonucleotides that hybridize to adjacent locations on a target sequence. Each carries a fluorophore from a FRET donor-acceptor pair. When both probes are bound to the target sequence, the fluorophores are brought in close proximity and allow FRET to occur. In 2012, Niederholtmeyer et al. performed successful real-time detection of transcriptional mRNA production during in vitro transcription and translation (ITT) reactions using two 15 bp probes that bind Cy3 and Cy5 fluorophores, respectively, from the sensitized fluorescence of Cy5[21]. Although this research showed mRNA production could be measured in real-time with FRET, other common, simplified RNA techniques have not been explored.

In this study, we created a real-time quantitative in vitro transcription (RT-IVT) assays for IVT-mRNA detection using "binary fluorescence quencher (BFQ)" probes using Chlorella virus DNA ligase (PBCV-1 DNA ligase) (Fig. 1). Here, we stabilized the signal from the binary probes and increased their universal application for multiple real-time quantitative detection. Compared with the binary probes designed by the Niederholtmeyer team, our BFQ probes changed the labeling site of the fluorophores and adjusted the detection signal from fluorescence sensitization to fluorescence quenching. Together, this system expanded the types of usable fluorophores and extended the detection instrument from a microplate reader to a multi-channel fixed-wavelength RT-PCR thermocycler, which allowed for simultaneous tracking of multiple promoter activities. In particular, the PBCV-1 DNA ligase was chosen to stabilize the FRET signal because of its unexpectedly high activation for RNA-splinted DNA ligation and easy availability[22–24]. Using this RT-IVT method, we developed an efficient assay for real-time activity monitoring of T7 RNA polymerase and *E.coli* RNA polymerase in unpurified solutions. We then used this technique to investigate the binding affinities of transcription regulators to their ligand or target DNA sequence. Altogether, we created an inexpensive multiplexed method to simultaneously measure real-time quantitative mRNA production, and these techniques could be utilized for future research to measure the affinity of transcriptional repressors to their target DNA sequence.

## Results

**RT-IVT method detection is based on binary fluorescence quencher probes and PBCV-1 DNA ligase.** We designed a real-

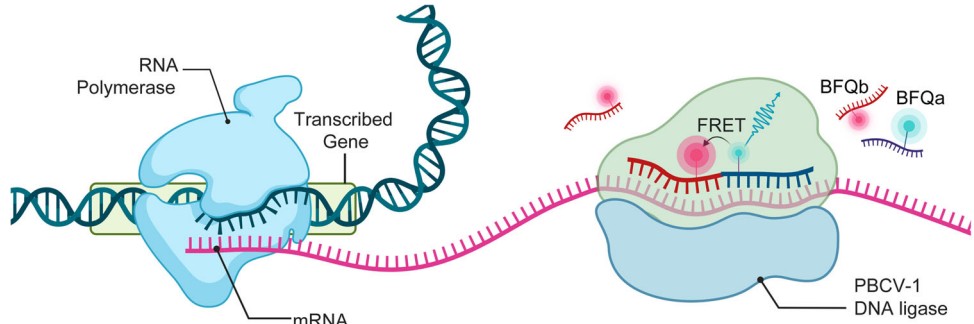

**Fig. 1 Schematic overview of the RT-IVT method for IVT-mRNA quantification.** mRNA with the target site for the BFQ probes is transcribed from a DNA template. BFQ probes (BFQa and BFQb) carrying either a donor fluorophore or quencher fluorophore are present in the IVT mixture and hybridize to the mRNA. This brings the donor and quencher fluorophores into close proximity so that FRET can occur. The FRET signal allows quantification of mRNA concentration in real-time. Figure was created with Biorender (www.bioender.com).

**Table 1 The fluorophore-labeled single-stranded DNA probes used in this study.**

| Probe | | Sequence& Fluorescent mark |
|---|---|---|
| BFQ1 | BFQ1a2 | 5′-pT/FAM/TTTTGGATGGTGGTTGACGGCG-3′ |
| | BFQ1a3 | 5′-pTT/FAM/TTTGGATGGTGGTTGACGGCG-3′ |
| | BFQ1a4 | 5′-pTTT/FAM/TTGGATGGTGGTTGACGGCG-3′ |
| | BFQ1a5 | 5′-pTTTT/FAM/TGGATGGTGGTTGACGGCG-3′ |
| | BFQ1b2 | 5′-TTGCCCCAGCAGGCGTTTT/BHQ1/TC-3′ |
| | BFQ1b3 | 5′-TTGCCCCAGCAGGCGTTT/BHQ1/TTC-3′ |
| | BFQ1b4 | 5′-TTGCCCCAGCAGGCGTT/BHQ1/TTTC-3′ |
| | BFQ1b5 | 5′-TTGCCCCAGCAGGCGT/BHQ1/TTTTC-3′ |
| | BFQ1b6 | 5′-TTGCCCCAGCAGGCG/BHQ1/TTTTTC-3′ |
| BFQ2 | BFQ2a2 | 5′-pC/VIC/TACGGCGTTTCACTTCTGAGTTCGGC-3′ |
| | BFQ2b3 | 5′-GGGAGACCCCACACTACCATCG/BHQ2/TCT-3′ |
| BFQ3 | BFQ3a2 | 5′-pT/ROX/TGCAGCAGCGGTCGGCAGCAGGTAT-3′ |
| | BFQ3b3 | 5′-TGGGCAGCGAGGAGCAGCA/BHQ2/TGC-3′ |
| Binary probes | | 5′-/Cy3/ACCAATGGGCTCAGT-3′ |
| | | 5′-GAGTCCTTCC/Cy5/ACGAT-3′ |

time quantitative in vitro transcription (RT-IVT) assay for IVT-mRNA quantification using a binary fluorescence quencher (BFQ) probe and the PBCV-1 DNA ligase. BFQ usually consists of two single-stranded DNA (ssDNA) molecular probes (BFQa and BFQb) that were modified by a donor fluorophore and a quencher fluorophore, separately. They were designed to totally sequence complementary on the same mRNA target at adjacent regions, so that FRET only occurred when both probes were hybridized to the same target. After hybridization, the PBCV-1 DNA ligase ligated the mRNA-splinted two DNA probes into one single DNA strand to stabilize the FRET signal. Therefore, the amount of fluorescence quenching detected during a transcription reaction was proportional to the mRNA yielded (Fig. 1).

In order to test the availability of BFQ and PBCV-1 DNA ligase in IVT experiments, we synthesized a pair of BFQ1 probes (BFQ1a2 modified by FAM on the second nucleotides of 5′ terminal, BFQ1b2 modified by BHQ1 on the second nucleotides of 3′ terminal) (Table 1). The corresponding complementary RNA (cRNA in Supplementary Table S1) and the purified PBCV-1 DNA ligase were then used to test their fluorescence quenching and RNA-splinted DNA ligation efficiency. Our results showed that the 500 nM BFQ1a2 and BFQ1b2 in the presence of equivalent cRNA quenched the fluorescence signal by 47.5%. While the fluorescence signal continued to decrease as the concentration of DNA ligase increased, but stabilized at 50 nM (70% fluorescence quenching) (Fig. 2a). We then investigated the ligation activity PBCV-1 DNA ligase from 0–200 nM for 500 nM BFQ1a2 and BFQ1b2 by using agarose gel electrophoresis, and the results are shown in Fig. 2b. We found that PBCV-1 DNA ligase can efficiently ligate ssDNA probe fragments splinted by unbroken complementary RNA strands[22], and our results indicate that 50 nM PBCV-1 DNA ligase could ligate 95% of the ssDNA in the system.

We then synthesized BFQ1a3–BFQ1a5 and BFQ1b3–BFQ1b6 to optimize the labeling site of donor and quencher fluorophores. The nucleotide sequence of the new probes was the same as BFQ1, but the labeling site of their fluorophores was adjusted (Table 1). All of the above probes were combined with each other to detect the fluorescence quenching rate after the equivalent amount of cRNA and 50 nM PBCV-1 DNA ligase were added. With the probes, the fluorescence quenching efficiency of BHQ1 with FAM had a noticeable position effect, while BFQ1a2 and BFQ1b3 had the highest FRET efficiency and reached 56.8% fluorescence quenching after cRNA addition and 79.4% after cRNA and PBCV-1 DNA ligase addition (Fig. 2c). Figure 2d shows that the BFQ1a2 and BFQ1b3 set had the largest amount of

fluorescence quenching. From these experiments, the BFQ1a2, BFQ1b3 probes and 50 nM PBCV-1 DNA ligase were selected for further experiments.

Subsequently, we continuously increased the concentration of the BFQ1 (BFQ1a2 and BFQ1b3) probe in the solution to estimate their detection range and elucidate the contribution of trackless motion to probe quenching. Surprisingly, the linear concentration of BFQ1 was between 0 and 1 μM (Fig. 2e), which is sufficient for the detection of IVT-mRNA production, which is generally less than 500 nM mRNA.

**Using RT-IVT to monitor transcription processes in real-time.**
A series of in vitro transcription (IVT) assays of the *E. coli* T7 RNA polymerase (T7 RNAp) were set up to demonstrate the real-time quantitative detection effect of our RT-IVT method. The reagent addition scheme for the experimental and negative control groups (equal volume of $H_2O$ instead of NTP mixtures) are described in the materials and methods section. After transcription reagent configuration with the 500 nM BFQ1 (BFQ1a2 and BFQ1b3) and 50 nM DNA ligase, the system was monitored at 37 °C in real-time by the RT-PCR thermocycler.

Figure 3a shows the raw data of the real-time fluorescence scan curves got T7 RNA polymerase with different concentrations of DNA template, where DNA template 1 includes T7 promoter and the BFQ1 binding region, and is described in Supplementary Table S1. In order to display the transcriptional activity more intuitively, the signal was corrected by the negative control minus the experimental group (signal correction) and redrawn in Fig. 3b. Relative initial velocity for the mRNA transcription of the 30 nM DNA template group was measured by fitting the slope for the reaction's linear region (generally the first 10% of the reaction), which is also marked in Fig. 3b. According to these results, the RT-IVT assays could monitor the IVT process in a continuously and with high sensitivity.

To compare the effectiveness of the BFQ probes, PBCV-1 DNA ligase and the reported binary probes[21] in IVT assays, as well as the DNA template[BFQ1&Binary probes], were synthesized to detect the T7 RNAp transcription process using a microplate reader. Here, the BFQ1 probes and PBCV-1 DNA ligase, BFQ1 probes only, and binary probes were analyzed separately (Table 1). Specifically, the DNA template[BFQ1&Binary probes] contained both BFQ1 and binary probe-tagged sites (Supplementary Table S1). As a result, the quenching detection signal of the BFQ1 probes in the IVT assays were approximately 8x higher than the sensitization signal of the binary probes. Furthermore, the addition of PBCV-1 DNA ligase increased the detection efficiency

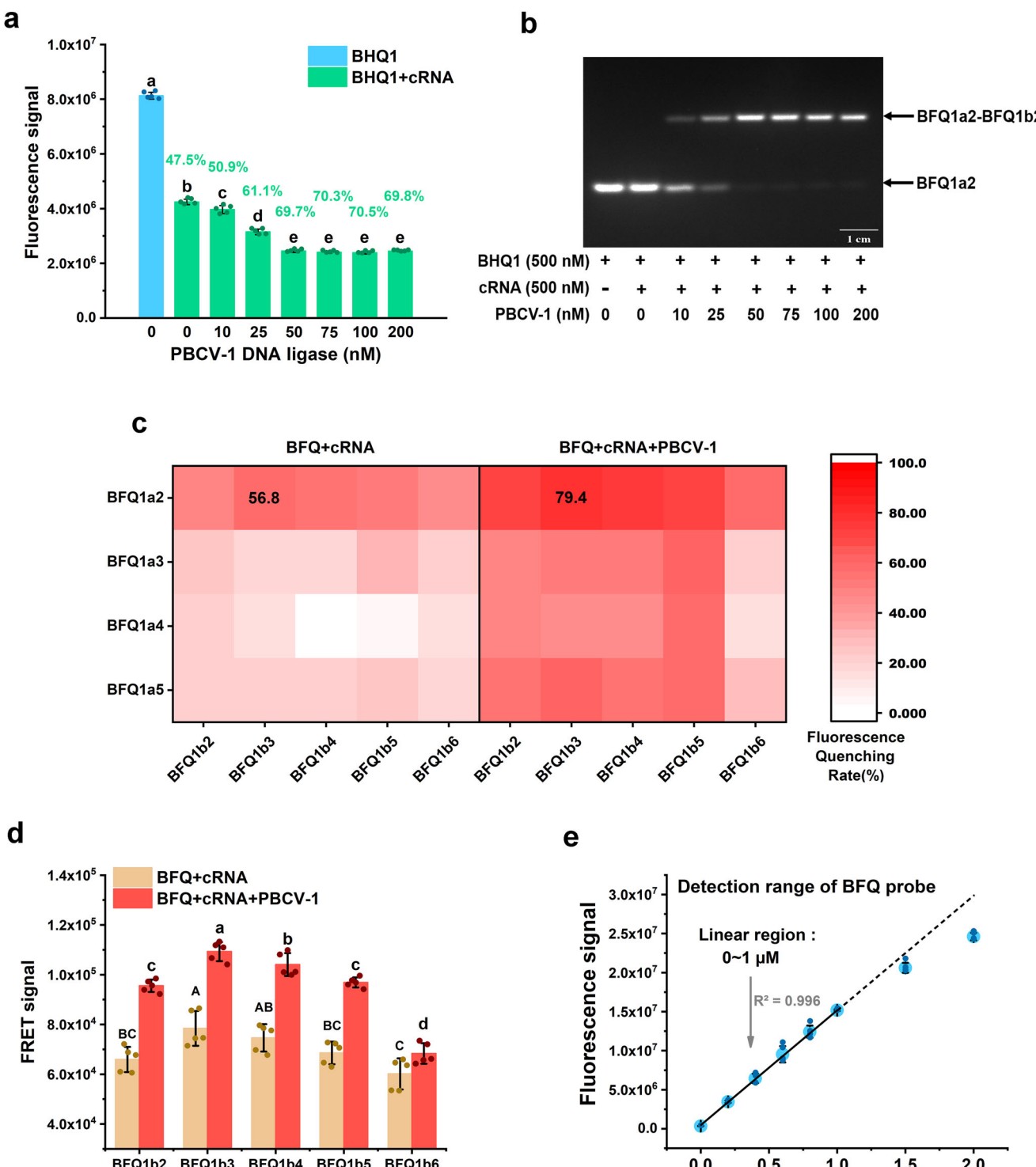

**Fig. 2 BFQ and PBCV-1 DNA ligase can be used for RNA quantification in IVT reactions. a** Addition of PBCV-1 DNA ligase facilitated FRET efficiency in the detection system. 500 nM BFQ1 (BFQ1a2 and BFQ1b2), 500 nM cRNA, and different concentrations of PBCV-1 DNA ligase were mixed in transcription buffer and tested using a RT-PCR thermocycler with its FAM scan channels. Fluorescence quenching rates were calculated by the fluorescence quenching value of each group divided by the fluorescence value of only the BFQ1 group and is displayed in the corresponding position of the picture. Different letters indicate significant differences among PBCV-1 DNA ligase concentrations according to Duncan's test ($p < 0.05$, $n = 5$). **b** The RNA-splinted DNA ligation assays for different concentration of PBCV-1 DNA ligase. **c** Confirmation of fluorophore labeling positions in the BFQ probes. The sequence and fluorophore labeling positions are shown in Table 1. The fluorescence quenching rate refers to the ratio of the fluorescence quenching value of each group to the fluorescence value of the corresponding BFQ1a probe and is displayed in the corresponding positions. **d** The FRET quenching signal of BFQ1a2 with BFQ1b2-BFQ1b6. Different letters indicate significant differences among PBCV-1 DNA ligase concentrations and fluorophores labeling positions adjusted according to Duncan's test ($p < 0.05$, $n = 5$). **e** Detection range identification of BFQ. Different concentrations of BFQ1 (BFQ1a2 and BFQ1b3) were mixed in transcription buffer and tested using a RT-PCR thermocycler. All of the values shown represented the mean ± standard deviation of the results from five independent experiments.

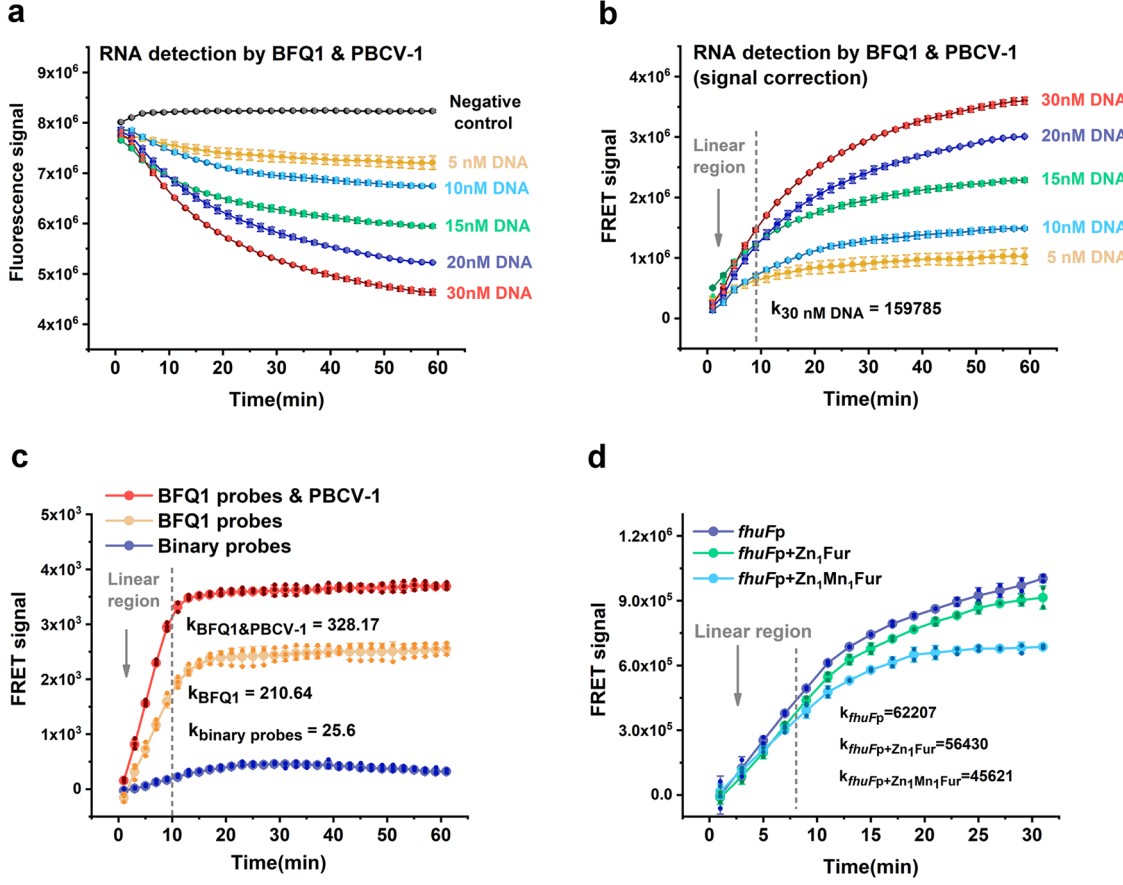

**Fig. 3 Real-time quantitative transcription detection assays for BFQ and PBCV-1 DNA ligase. a, b** The raw and signal corrected data for T7 RNAp transcription activity. 200 nM T7 RNA polymerase, 5–30 nM DNA template 1, 250 μM NTP mixture, 500 nM BFQ1 (BFQ1a2 and BFQ1b3) and 50 nM PBCV-1 DNA ligase were mixed in transcription buffer and immediately tested in a RT-PCR thermocycler using its FAM scan channels. The initial transcription velocity for the 30 nM DNA template group was measured by fitting the slope for the 1–9 min reaction region. Values are mean ± standard deviation of two repeats. **c** Comparison of the binary probes and RT-IVT methods. 200 nM T7 RNA polymerase, 30 nM DNA template$^{BFQ1\&Binary\ probes}$, 250 μM NTP mixture, and 500 nM binary probes or 500 nM BFQ1 and 50 nM PBCV-1 DNA ligase (added as required) were mixed in transcription buffer and tested in a microplate reader with FAM fluorescence (excitation 494 nm, emission 522 nm) and Cy3/Cy5 sensitized fluorescence (540 nm, 680 nm) separately. Values are mean ± standard deviation of three repeats. **d** The RT-IVT method can be used for detection of transcriptional repression activity for Fur. 250 nM *E. coli* RNA polymerase holoenzyme, 30 nM DNA template$^{fhuFp}$, 250 μM NTP mixture, 250 nM BFQ1 (BFQ1a2 and BFQ1b3), 50 nM PBCV-1 DNA ligase and 500 nM $Zn_1Fur$ or $Zn_1Mn_1Fur$ were mixed in transcription buffer and tested immediately in a RT-PCR thermocycler using its FAM scan channels. Values are mean ± standard deviation of two repeats.

of the BFQ1 probes by 1.6x, and improved the signal-to-noise ratio for BFQ1 and binary probes by 3.9x and 6.2x, respectively (61.5 for DNA ligase added, 15.6 for BFQ1 probes only and 10 for binary probes), which made their efficiency 12.8x higher than the binary probes (Fig. 3c).

The the RT-IVT method was then used to verify the effect of ligand molecules on the activity of transcription regulators. In *E. coli*, the regulation of intracellular iron homeostasis is dependent on the ferric uptake regulator (Fur) protein[25]. Under iron sufficient conditions, the metal ion saturated $Zn_1Fe_1Fur$ (or $Zn_1Mn_1Fur$) combines with the operator site of a target promoter, which blocks the binding of the RNA polymerase and inhibits iron-uptake gene transcription. While under iron-limited conditions, $Zn_1Fur$ dissociates from the target DNA sequence and relieves iron-uptake gene repression[26]. Additionally, the *fhuF* promoter contains multiple overlapping Fur protein binding sites[27]. In this experiment, we synthesized a *fhuF* promoter-BFQ1 transcription model (DNA template $^{fhuFp}$ in Supplementary Table S1) and tested Fur's regulated transcriptional activity using the *E.coli* RNA polymerase holoenzyme. Figure 3d shows the real-time fluorescence scan curves detected

by 250 nM BFQ1 and 50 nM DNA ligase (after signal correction). As expected, $Zn_1Fur$ and $Zn_1Mn_1Fur$ inhibited the transcription rate of the *fhuF* promoter by 9.3% and 26.7%, respectively, which was consistent with previous detection using radioactively tagged IVT methods[28]. These data suggest that the RT-IVT assays can monitor transcription processes and detect transcription regulator activity in real-time.

Since ATP concentration, NaCl, and metal ion content can affect the activity of RNA polymerase and PBCV-1 DNA ligase, we further investigated the effect of buffer conditions on transcription detection using these RT-IVT methods. The standard reaction system was 200 nM T7 RNA polymerase, 30 nM DNA 1 template, 500 nM BFQ1, 50 nM PBCV-1 DNA ligase, 40 mM Tris pH 7.0, 30 mM $MgCl_2$, 2.5 mM DTT and 5% glycerol. Except for the NTP optimization experiment, the NTP concentration was 250 μM for the other groups. The relative initial transcription velocity of T7 RNA polymerase were calculated with the different buffer conditions and displayed in Fig. 4a–f. From these results, the optimum reaction conditions for the NTP mixture were 200–300 μM with pH 7.0 using 200 nM T7 RNA polymerase and 30 nM DNA template for transcription. High concentrations of NaCl

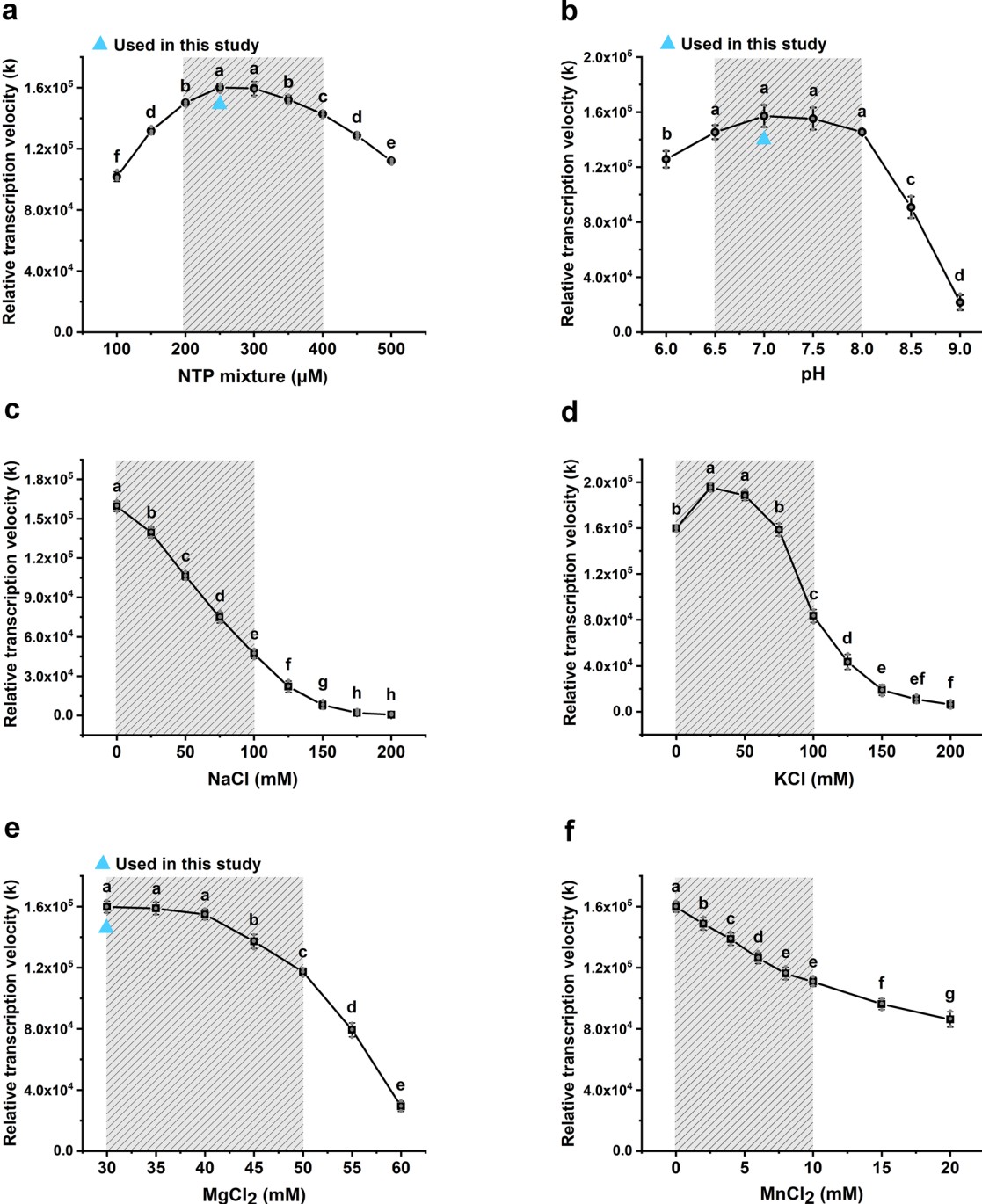

**Fig. 4 Effects of reaction buffers on RT-IVT assays.** The relative initial transcription velocity of T7 RNA polymerase at different NTP (**a**), pH (**b**), NaCl (**c**), KCl (**d**), $MgCl_2$ (**e**), and $MnCl_2$ (**f**) concentrations. The shaded areas represent the common concentration range for each ion, and triangles represent the ion concentrations used. The values shown were the mean ± standard deviation of three repeats. Different letters indicate significant differences among NTP, pH, or ion treatments according to Duncan's test ($p < 0.05$, $n = 3$).

significantly inhibited transcription, while 0–75 mM KCl promoted the response (Duncan's test, $p < 0.05$, $n = 3$). Furthermore, the tolerance concentration of the RT-IVT assays for $MgCl_2$ and $MnCl_2$ were 30–45 mM and 0–4 mM, which affected less than 10% of the transcription rate (Fig. 4a–f).

**Multivariate and quantitative real-time detection of IVT-mRNA production.** After investigating the effect of different reagents in buffer, we used our multiplex real-time quantitative in vitro transcription (RT-IVT) method to simultaneously detect

different mRNAs in the same transcription system with BFQs that were specific for different targets that were labeled with different colored fluorophores and quenchers, which are described in Fig. 5a.

To accomplish this, we synthesized three different BFQs where their sequences and colors corresponded to their complementary DNA (cDNA). Together, we synthesized them with their transcription models: (1) BFQ1 (BFQ1a2 and BFQ1b3) modified by FAM/BHQ1 fluorophores and targeted to cDNA 1 and DNA template 1, (2) BFQ2 (BFQ2a2 and BFQ2b3) modified by VIC/BHQ2 fluorophores and targeted to cDNA 2 and DNA template

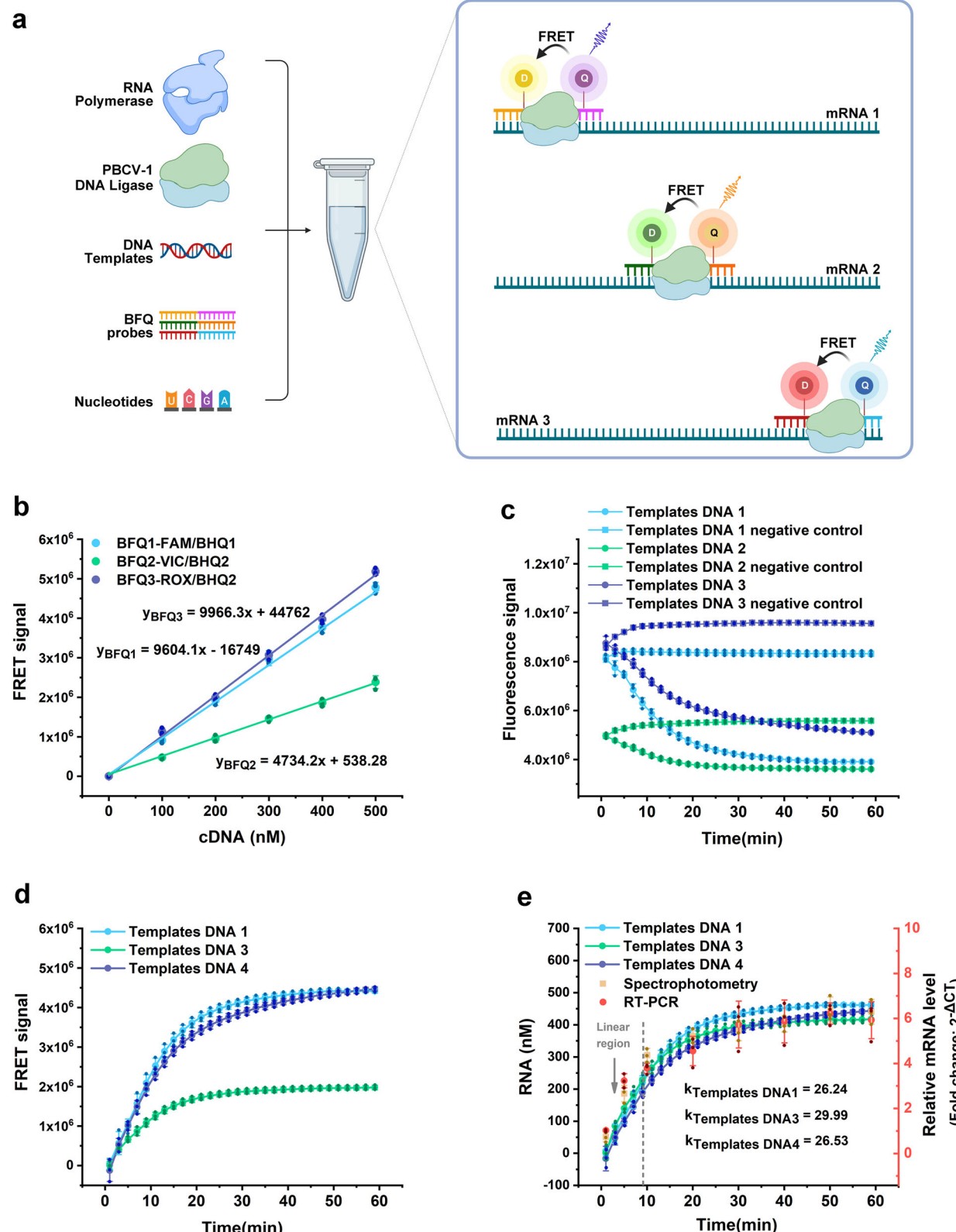

2, and (3) BFQ3 (BFQ3a2 and BFQ3b3) modified by ROX/BHQ2 fluorophores and targeted to cDNA 3 and DNA template 3 (Table 1 and Supplementary Table S1). The above DNA templates contained the same T7 promoter fused to different probe recognition sequences. The BFQs were then calibrated through hybridization with gradually increasing concentrations of the corresponding cDNAs, respectively, to test their target-signal

standard curves (Fig. 5b). During transcription, the three different DNA templates (DNA 1, DNA 2, and DNA 3), three different BFQ pairs (BFQ1, BFQ2, and BFQ3) with T7 RNAp, and PBCV-1 DNA ligase were mixed and then analyzed in the RT-PCR thermocycler at 37 °C by its FAM/VIC/ROX scan channels.

Figure 5c, d show the raw data and the relative quantitative (the negative control signal data was corrected) real-time

**Fig. 5 Using BFQs and PBCV-1 DNA ligase for multivariate and quantitative detection of IVT-mRNA production. a** Principle of multiplex RT-IVT method. Figure was created with Biorender (www.bioender.com). **b** The target-signal standard curves for BFQ1, BFQ2 and BFQ3. The FRET signal represents the fluorescence difference between each group and the 0 nM cDNA group (control). **c–e** The raw, relative quantitative corrected signal and absolute quantitative corrected signal results for the three DNA templates with their transcriptional activities. 200 nM T7 RNA polymerase; 30 nM DNA template 1, DNA 2 and DNA 3; 500 μM NTP mixture; 500 nM BFQ1, BFQ2 and BFQ3; and 50 nM PBCV-1 DNA ligase were mixed in standard reaction buffer and immediately and simultaneously tested in a RT-PCR thermocycler at 37 °C using its FAM/VIC/ROX scan channels. The orange square and red circles in Fig. 5e represent the transcriptional activity results for DNA template 1 tested by the spectrophotometric and the RT-PCR methods. Between the groups, the relative quantitative detection results for the RT-PCR method corresponded to the right, red coordinate axis. The values represent the mean ± standard deviation of three repeats.

---

**Table 2 The collections of the current popular IVT-mRNA detection methods.**

| Detection methods | Duration | Price/reaction[a] | Detection components | Instrument | Origins |
|---|---|---|---|---|---|
| RT-IVT | 3 h (real-time) | 3.804 or 0.104[b] | BFQ probes (Sangon Biotech) PBCV-1 DNA ligase (NEB) | RT-PCR thermocycler or microplate reader | This study |
| Spectrophotometry | 5 h | 4.458 | RNA purification kit (TaKaRa) | Spectrophotometer | (11) |
| RT-PCR | 8 h | 14.117 | RNA purification kit (TaKaRa) cDNA Synthesis kit (TaKaRa) SYBR Premix PCR kit (TaKaRa) | RT-PCR thermocycler | (10) |
| Radioactively tagged nucleotides replacement | 12 h | 14.860 | [$\alpha$-$^{32}$P] UTP mixed with ATP, CTP, and GTP (PerkinElmer) Denaturing gel electrophoresis | Radiological safety laboratory and scintillation counter | (12) |

[a]The cost only applies to the detection components in the reaction and the price was transferred by rate calculation of RMB to USD.
[b]The price of 3.804 was calculated using BFQ probes added to PBCV-1 DNA ligase purchased from New England Biolabs (NEB) corporation, while the price of 0.104 was calculated using BFQ probes added to PBCV-1 DNA ligase purified in our laboratory.

---

fluorescence scan curves detected by three channels. Figure 5e showed the absolute quantitative real-time mRNA production curves for the three transcriptions after calibration with their target-signal standard curves for the probes. Our results showed that the transcription rates of the three DNA templates were very similar (DNA template $1 = 26.24$ nM min$^{-1}$ and 64.29 nt s$^{-1}$, DNA template $2 = 29.99$ nM min$^{-1}$ and 71.48 nt s$^{-1}$, DNA template $3 = 26.53$ nM min$^{-1}$ and 66.77 nt s$^{-1}$), which was consistent with previous reports where the in vivo transcription rate of T7 RNAp is approximately 200 nt s$^{-1}$ and is roughly 5x higher than the in vitro transcription rate[29,30]. Therefore, we were able to use this RT-IVT method with differently labeled BFQ probes for the simultaneous detection of different transcriptional activity with multiple promoters in a homogeneous system.

To determine the validity of our RT-IVT method, mRNA production was then confirmed by the spectrophotometry and the RT-PCR method with the DNA template 1 group. From our results, the transcription rates of the DNA 1 templates was 28.178 nM min$^{-1}$ using spectrophotometry, which was similar with our RT-IVT method (26.24 nM min$^{-1}$). As a relative quantitative detection method, the initial slope for the RT-PCR monitoring curve was also similar to our RT-IVT method (Fig. 5e).

Following these experiments, we compared the time and economic cost of our RT-IVT method with current, popular IVT-mRNA detection methods, which are listed in Table 2. As a real-time quantitative detection method, the RT-IVT method has a shorter detection duration, is more inexpensive, and more user-friendly, which is more conducive to the application of IVT technology.

**Rapid detection of DNA-binding coefficients for transcriptional inhibitors using RT-IVT.** DNA binding is a critical step for regulatory proteins to function as transcription activators or inhibitors. The equilibrium dissociation constant ($K_D$ values) between regulator and box DNA is an important index to

evaluate the strength of regulation. In these experiments, we used the RT-IVT method to determine the inhibitor-DNA binding $K_D$ values to show its broad applications. Figure 6a shows a scheme with how transcriptional DNA templates are synthesized and include the T7 promoter followed by the box DNA sequence for inhibitors and a BFQ targeted region. During the RT-IVT assays, we added increasing concentrations of inhibitors to detect their corresponding transcription velocities and drew a curve with the inhibitor concentration on the y-axis and the corresponding transcriptional inhibition velocity on the x-axis. The inhibitor concentration corresponded to half of the maximum transcriptional inhibition speed and represents the $K_D$ value of the inhibitor with their box DNA.

The ferric uptake regulator (Fur) protein and its classical DNA-binding sequences (Fur box DNA: GATAATGATAATGATAAT-GATAATGATAATGA) were used to test this method[31]. The synthesized transcriptional DNA template$^{Fur}$ sequences are also shown in Supplementary Table S1, and the RT-IVT results for BFQ1 and PBCV-1 DNA ligase are shown in Fig. 6b. The final results indicated that the $K_D$ value for Fur binding to Fur box DNA was $0.35 \pm 0.12$ μM (Fig. 6c), which was similar to the FP method ($0.56 \pm 0.03$ μM)[28].

## Discussion

IVT assays are a classic procedure to study gene expression, and the frequency of cited research using them has been rapidly increasing over the last decade. However, the current detection methods are expensive and do not allow for real-time tracking of transcription. In this study, we developed a RT-IVT method to rapidly monitor in vitro transcription processes in homogeneous solutions using binary fluorescence quencher (BFQ) probes and PBCV-1 DNA ligase. Compared to classical MB and the binary probes[21], our method provides better detection sensitivity and stability that is not confounded by DNA-binding proteins, such as RNA polymerase. More importantly, this method enables the

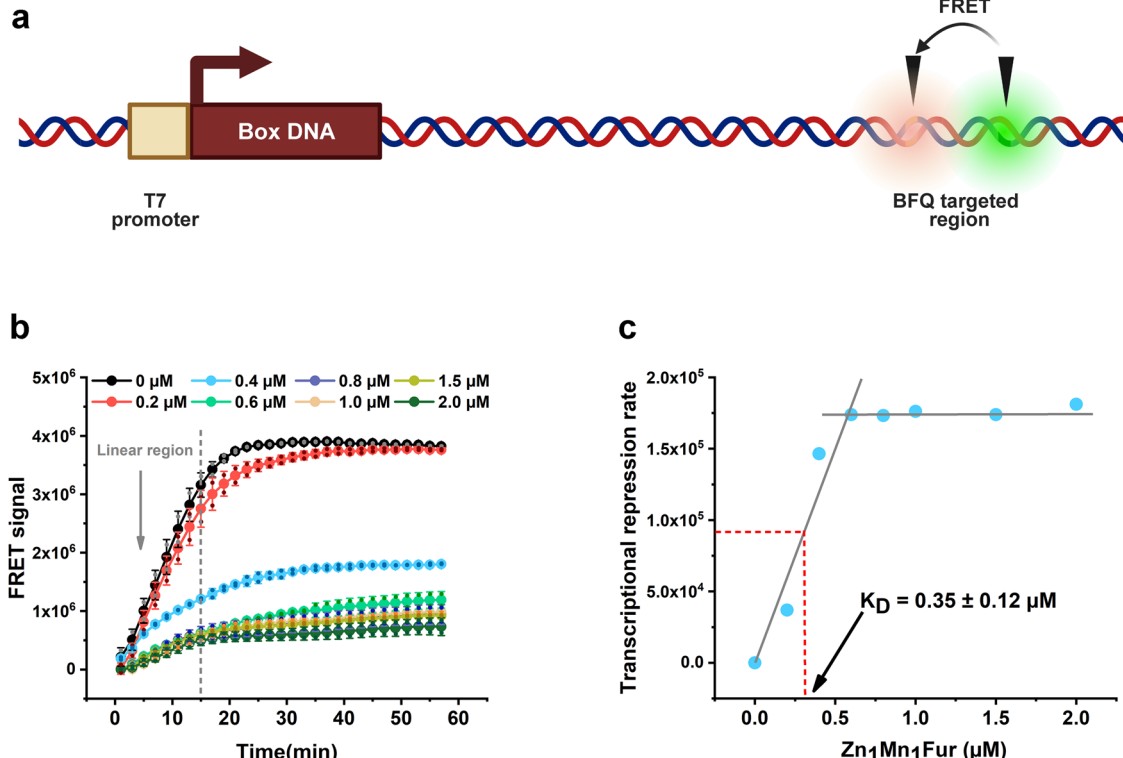

**Fig. 6 Using RT-IVT assays to detect the DNA-binding coefficient of transcriptional inhibitor. a** Principles for DNA template design. Figure was created with Biorender (www.bioender.com). **b** The relative quantitative results (signal corrected) for the DNA template$^{Fur}$ after $Zn_1Mn_1Fur$ was added. The reaction configuration and detection conditions are identical to those shown in Fig. 3a, except 30 nM DNA template$^{Fur}$ replaced DNA template 1, and 0–2 μM $Zn_1Mn_1Fur$ was added. **c** The $K_D$ value for Fur binding to the Fur box DNA. The initial transcription velocity for the 0–2 μM $Zn_1Mn_1Fur$ added groups were measured by fitting the slope of the 1–15 min reaction region of Fig. 6b. The transcriptional repression rates were calculated using the initial transcription velocity of 0 μM $Zn_1Mn_1Fur$ group subtracted by the corresponding groups. The $Zn_1Mn_1Fur$ concentration corresponds to half of the maximum transcriptional inhibition rates and represents the $K_D$ value of $Zn_1Mn_1Fur$ with the Fur box DNA. The values represent the mean ± standard deviation of $n = 2$ replicates.

simultaneous detection of transcriptional activity for multiple promoters in one system.

Since RT-IVT detection only focuses on the donor fluorophore quenching and not the sensitization of an acceptor fluorophore, this process greatly increases the sensitivity of the detection signal and allows for more choices with labeling fluorophores. Additionally, commonly used donor/quencher fluorophore pairs, such as FAM/BHQ1, FAM/TAMRA, ROX/BHQ2, VIC/BHQ2, and CY5/BHQ3, are all ideal choices for BFQ pair labeling[32]. A more important advantage with quenched-signal detection is to extend the detection instrument from an adjustable wavelength microplate reader to a fixed channel RT-PCR thermocycler. With the outbreak of COVID-19, the RT-PCR thermocycler has become popular around the world and brings convenience to the implementation of the new technology. In particular, almost all RT-PCR thermocyclers can support fixed-wavelength multi-channel detection (usually 2–5 channels), which is beneficial to analyze the transcriptional activity of multi-promoters or bidirectional-promoters. Additionally, the multi-channel method can also be applied to test RNA integrity by designing multiple pairs of target probes with a long fragment of RNA, which is very important for the production of RNA vaccines[7,8].

When multiple promote activities are detected, the productions can be distinguished from one another by using BFQs that are specific for different mRNAs and are labeled with differently colored fluorophores. Here, we investigated the simultaneous real-time quantitative detection of transcription activities of three promoters in a homogeneous transcriptional system. For more

fluorophore combinations, the following principles could be considered: non-overlapping fluorescence detection channels, achievable instrument detection ranges, and the reasonable synthesis cost of fluorophore-labeled nucleic acid. Together, these could be explored in the future to improve RT-IVT multi-promoter activities.

With the RT-IVT method, a suitable target site that accommodates both nucleic acid probes is the key to their success[33]. In the first approach, we could analyze the original nucleic acid sequence after the promoter, select a target region, and synthesize probes according to this sequence. The new probes had specific properties that included: (1) 20–30 nt nucleotides in length, (2) Tm values between 60 to 80 °C, (3) GC content between 0.5 to 0.7, and (4) minimal hairpin structure and dimer formation. In the second approach, an available probe hybridization sequence could be added in the IVT DNA template that was approximately 100–2000 nt away from the promoter site. The second method could omit the probe design stage, which expedites screening since new probes do not need to be synthesized each time, which is more cost-effective.

From our results, it is worth noting that the IVT rate splits in the earlier stage of the reaction, so it is necessary to ensure timely detection during the initial reaction. Our method scheme configures the reaction reagents and the NTP mixture separately, and the NTP mixture is only added after the testing instrument has been properly set up. Additionally, temperature changes can significantly affect the fluorescence detection signal, so experiments should be carried out under a constant temperature, such as 37 °C.

In general, the equilibrium dissociation constant ($K_D$ values) between regulator and box DNA can be accurately detected using fluorescence polarization spectroscopy (FP), microscale thermophoresis (MST) or isothermal titration calorimetry (ITC)[34–36]. However, $K_D$ value determination is relatively difficult. For example, ITC is only suitable for detecting significant thermal changes, yet uniform particle distribution is needed for FP and MST. Still, a majority of proteins bind DNA with insignificant changes in heat, and many proteins tend to randomly polymerize in solution, even after surfactants are added. In this study, we extended the application of the RT-IVT method to detect inhibitor-DNA binding $K_D$ values, which could partially tolerate the random aggregation of regulatory proteins.

In summary, we developed a highly sensitive, extremely selective, non-radioactive, easily detectable and economical method to quantitatively detect IVT-mRNA production in real-time. From reagent preparation to data analysis, the whole experimental setup could be completed within 3 h. The development of this method has significantly lowered the threshold to use IVT technology, which will further promote its application in life science research.

## Methods

**Materials**. The fluorophore-labeled single-stranded DNA probes (BFQ1, BFQ2, BFQ3 and binary probes) and oligonucleotides used as primers and cDNAs were synthesized by Sangon Biotech (Sangon, Shanghai, China) and sequences were listed in Table 1, Supplementary Tables S1 and S2. DNA templates for the IVT reactions were amplified using pET22b plasmids or *E. coli* MG1655 genome with PCR using Phusion High Fidelity DNA polymerase (New England Biolabs, Ipswich, MA, USA) and their gene-specific primers (Supplementary Table S2), which introduced the probe target sites in the 3′ extension primers. After amplification, PCR products were purified and their concentrations were determined with a Nanodrop spectrophotometer (Biofuture, K5600, Beijing, China). All nucleic acid samples were diluted to appropriate concentration with RNA-free $H_2O$ and stored at −20 °C for later use.

The T7 RNA polymerase, *E.coli* RNA polymerase holoenzyme, and NTP mixture were also purchased from New England Biolabs and Sangon Biotech. Other biochemical reagents were purchased from Sigma-Aldrich (Saint Louis, MO, USA).

The wild-type PBCV-1 DNA ligase gene[37] was synthesized by the Beijing Genomics institution and subcloned into the NdeI/XhoI sites of the pET29b vector with a C-terminal His-tag. The protein was overexpressed in *E. coli* BL21 (DE3) grown in Luria-Bertani (LB) medium supplemented with 0.1% glucose and 0.1% $MgCl_2$, and induced by 0.4 mM Isopropyl b-D-1-thiogalacto-pyranoside (IPTG) at 16 °C. After 20 h of induction, the cells were harvested in lysis buffer (20 mM Tris–HCl, pH 8.0, 200 mM KCl, 5 mM β-mercaptoethanol, 10% glycerole, and 1 mM PMSF) and lysed by sonication. The lysate was centrifuged at 28,500 × *g* for 50 min, and the supernatant was then loaded onto a Ni-NTA column (GE Healthcare, Pittsburgh, PA, USA) for affinity chromatography. After elution from the column in the elution buffer (20 mM Tris–HCl, pH 8.0, 200 mM KCl, 5 mM β-mercaptoethanol, 10% glycerole and 250 mM imidazole), the sample was purified to >95% homogeneity by ion exchange chromatography (Source Q, GE Healthcare) using AKTA fast protein liquid chromatograph (FPLC). The purified protein was confirmed by HPLC-mass spectrometry, and then diluted in storage buffer (20 mM Tris–HCl, pH 8.0, 100 mM KCl, 2 mM DTT, and 50% glycerole) and stored at −80 °C.

The wild-type Fur protein was purified from *E. coli* BL21 that harbored the pGL01-*fur* plasmid, as described in the literature[28], using a Ni-NTA column for affinity chromatography and Superdex 200 (GE Healthcare) for size-exclusion chromatography. The samples were then dialyzed with buffer A (20 mM Tris–HCl, 100 mM NaCl, 50 mM EDTA, pH 6.8) or buffer B (20 mM Tris–HCl, 100 mM NaCl, 1 mM $MnCl_2$, pH 6.8) for $Zn_1Fur$ or $Zn_1Mn_1Fur$ preparation[38]. The protein purity was then analyzed using SDS-PAGE for accuracy and was concentration determined using a Nanodrop spectrophotometer.

**RNA-splinted DNA ligation assays**. 500 nM FAM labeled BFQ1a2 with 500 nM unlabeled BFQ1b2 (5′-TTGCCCCAGCAGGCGTTTTTC-3′), 500 nM cRNA, and different concentrations of PBCV-1 DNA ligase were mixed in a standard reaction buffer at 37 °C for 5 min. The samples were then treated with RNaseH at 37 °C for 1 h and tested using 6% agarose gel electrophoresis. The gel was imaged with FAM fluorescence using a Typhoon Scanner (GE Healthcare) and Imagequant software (GE Healthcare).

**IVT setup and fluorescence measurement**. For IVT assays, 200 nM T7 RNAp or 250 nM *E.coli* RNAp and 5-30 nM DNA template were mixed in transcription buffer (40 mM Tris pH 7.0, 30 mM $MgCl_2$, 2.5 mM DTT and 5% glycerol) incubated for 10 min at 37 °C for open complex formation. The 250-500 nM different probes, 50 nM PBCV-1 DNA ligase, and $Zn_1Fur$ or $Zn_1Mn_1Fur$ were then added as required. A 250 µM NTP mixture was added and the mixture was then immediately placed in a RT-PCR thermocycler (Analytik Jena AG, qTOWER3G, Jena, Germany) or microplate reader (Biotek, SynergyMx, VT, USA) for fluorescence detection (37 °C). The reaction volume was 20 µL, which was placed into 200 µL Eppendorf tubes. We measured FAM fluorescence (excitation 494 nm, emission 522 nm), VIC fluorescence (538 nm and 554 nm), ROX fluorescence (585 nm and 605 nm), and Cy3/Cy5 FRET sensitized fluorescence (540 nm, 680 nm) every 2 min for 1 h as needed[32].

For the negative control, we examined the effects of different combinations of reagents, such as the omission of DNA templates, NTPs, RNA polymerase, and addition of 5% SDS or 20 mM EDTA, respectively. As a result, missing DNA templates or NTPs were the best combination (Supplementary Fig. S1). Since DNA templates need to be added first to form a complex with RNA polymerase, we chose to omit the NTP mixture with an equal volume of RNase free $H_2O$ to act as the negative control.

For the probe standard curves, 500 nM BFQ pairs (BFQ1, BFQ2 or BFQ3), 50 nM PBCV-1 DNA ligase, and 0-500 nM cDNA were mixed in transcription buffer at 37 °C for 5 min and separately tested in the RT-PCR thermocycler using its FAM, VIC, or ROX scan channels.

**RNA purification, spectrophotometry, and RT-PCR**. The IVT systems reacted at different times and the mRNA produced was then purified with the MiniBEST Universal RNA Extraction Kit (TaKaRa, Dalian, China) according to the manufacturer's instructions. For spectrophotometric detection of RNA concentrations, the RNA samples were diluted 250-fold in TE buffer (10 mM Tris–HCl, 1 mM EDTA, pH 8.0) and the absorbance of $OD_{260}$ was detected using a spectrophotometer (METASH, UV-6000PC, Shanghai, China). For calculation, RNA concentration ($\mu g \, \mu L^{-1}$) = $OD_{260}$ × dilution factor × 40/1000[10].

For RT-PCR detection of RNA concentration, the RNA samples were reverse transcribed into cDNA using the PrimeScript RT reagent Kit (TaKaRa) and stored at −20 °C. After they were diluted 10,000-fold in RNase free $H_2O$, and 2 µL of sample were analyzed in 20 µL of reaction reagents from the SYBR Premix Ex Taq™ II Kit (TaKaRa) in the Quantitative Real-Time PCR System. Amplified primers are shown in Supplementary Table S2 and the relative transcript abundance was calculated using the $2^{-\triangle Ct}$ method[11,21].

**Statistics and reproducibility**. Data were tested for the homogeneity of variance, and then checked for the normality. Data that did not meet the requirements were log transformed to improve normality and homogeneity. One-way analysis of variance (ANOVA) followed by Duncan's multiple comparison were used to test the differences among treatments, which were performed at α = 0.05. The number of replicates and error bars have been defined in the figure legends.

**Reporting summary**. Further information on research design is available in the Nature Portfolio Reporting Summary linked to this article.

## Data availability

All genomic sequences used in this study are publicly available and were downloaded from the National Center for Biotechnology Information (https://www.ncbi.nlm.nih.gov/). Sequence of nucleic acid probes, DNA templates, and primers are described in Table 1, Supplementary Tables S1 and S2. All source data presented in figures have be uploaded as Supplementary Data 1. Any remaining information can be obtained from the corresponding author upon reasonable request.

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

## Acknowledgements

We would like to thank Dr. Joseph Elliot at the University of Kansas for his assistance with English language and grammatical editing of the manuscript. This work was supported by the National Natural Science Foundation of China [31970043], the China Postdoctoral Science Foundation [2021M701996], the Youth Foundation of Shandong Natural Science Foundation [ZR202102230675], and the Shandong Province Postdoctoral Innovation Project Foundation [202101007].

## Author contributions

L.G. and K.Z. designed the study. F.Z., Y.W., and X.W. completed the experimental operation and performed date analysis. H.D. and M.C. purified the PBCV-1 DNA ligase. H.W. standardized figures. N.D., W.H., and L.G. critiqued and revised the English writing and discussion of this manuscript. F.Z. wrote the manuscript with contributions from all co-authors.

## Competing interests

The authors declare no competing interests.
