## [Peer Review File · Communications Biology]

Reviewers' comments:

Reviewer #1 (Remarks to the Author):

The manuscript 'A method for multiplex real-time quantification of in vitro transcriptional mRNA production (RT-IVT)' reports an advanced assay for real time monitoring of mRNA generation. The method is based on 'Real-Time mRNA Measurement during an in Vitro Transcription and Translation Reaction Using Binary Probes' published in ACS Synthetic Biology, 2013. The authors improved its signal to noise ratio by replacing binary probes with fluorescence quenchers. This enabled 3-plex mRNA detection in real time. Multiplex assays bear many advantages so that this manuscript is meaningful in practical application.

In this context, I have 2 questions on the method.

1. This manuscript describes detailed optimization of assay protocol. I found that the condition for negative control group is unique. In addition to the omission of DNA templates, NTP in the reagents was eliminated. I could not find the explanation for the removal of NTP to get the baseline of the assay. Did the NTP in the reagent interferes signal? If so, the stability of baseline might be unstable in real IVT condition.
2. To deviate quantitative FRET signal, signal of negative control was measure in each assay condition in Fig 3 and 5. If the baseline measurement need to be accompanied, the benefit from multiplex assay is quite damaged. Why does the baseline signal appear changing even without target DNA? Is it possible to set a fixed baseline for final assay condition?

Reviewer #2 (Remarks to the Author):

Brief summary of the manuscript:

Overall impression of the work:

The manuscript by Zhang et al., entitled: "A method for multiplex real-time quantification of in vitro transcriptional mRNA production (RT-IVT)" describes a method that uses binary fluorescence quencher (BFQ) probes and a DNA ligase for the real time monitoring of the produced IVT-mRNA. The method is well-structured, innovative and very interesting. To strengthen the translational potency of this study, I would suggest as follows:

Specific comments:

Comment 1: Please proof-read again and make the appropriate language corrections.

Line 46: realized  conducted

Line 57: please use the abbreviation IVT-mRNA for in vitro transcribed messenger RNA

Line 60: pushing IVT trials  placing IVT-mRNA trials

Line 69: However, these methods all need  However, all these methods need

Line 71: for trace  to trace or for tracing

Line 90: delete an

Line 92: correct realize

Line 94: transcribed mRNA  IVT-mRNA

Line 94: a "binary fluorescence quencher (BFQ)" probes  delete "a"

Line 97: Rephrase "Which..." it is not gram marly correct

Comment 2: The results section must be more compact and this text (Line 109-129) should be relocated at the Methods section and the Discussion section. Perhaps, the authors could consider re-writing the results section.

Comment 3: Despite the advantage of the real time task, comparison between this method and more conventional approaches like spectrophotometry, or reverse transcription would highlight the importance of this method.

Reviewer #3 (Remarks to the Author):

In this manuscript, the authors have developed a new method RT-IVT for the real-time measurements of mRNAs in the IVT reaction. Their method leverages the fluorescence quenchers & DNA ligase and uses the FRET and RNA-splinted DNA ligation methods. By using this method, authors have characterized RNA polymerases from bacteria and phages. While the manuscript presents a method improvement; however, I have some reservations. please check my comments below:

Major/minor comments.

1. I recommend authors use a uniform font type and size across all figures.
2. Figure legends are too abstract or method oriented. For example Fig2a, the authors have shown a bar graph representing relative flou. ins. values. What do the error bars represent? Have authors performed any statistical tests to check if the decline in flou. values are statistically significant over increasing PBCV-1 concentrations? Please resolve FL-related issues across all figure legends.
3. Panel B, why do the first two rows have two samples with zero PBCV-1 concentrations? if there is any rationale, please mention it in the figure legends.
4. Fig 2D, why do authors compute the statistical significance for a few comparisons, not all?
5. I recommend that authors avoid adding methodological details in the figure legends but restrict to the panel explanation.
6. Fig 3 and 4, is it one sample? or is it experiment done as a single replicate? if not, please provide the normalized, scaled data of all replicated alongside the statistical test (ANCOVA?).
7. Authors have made interesting comments in the abstract/manuscript, i.e., their method is cheaper than other methods. I recommend authors revisit this claim and do provide substantial evidence (suppl. info) for this.

Response to Reviewer #1

The manuscript 'A method for multiplex real-time quantification of *in vitro* transcriptional mRNA production (RT-IVT)' reports an advanced assay for real time monitoring of mRNA generation. The method is based on 'Real-Time mRNA Measurement during an *in Vitro* Transcription and Translation Reaction Using Binary Probes' published in *ACS Synthetic Biology*, 2013. The authors improved its signal to noise ratio by replacing binary probes with fluorescence quenchers. This enabled 3-plex mRNA detection in real time. Multiplex assays bear many advantages so that this manuscript is meaningful in practical application.

In this context, I have 2 questions on the method.

1. This manuscript describes detailed optimization of assay protocol. I found that the condition for negative control group is unique. In addition to the omission of DNA templates, NTP in the reagents was eliminated. I could not find the explanation for the removal of NTP to get the baseline of the assay. Did the NTP in the reagent interferes signal? If so, the stability of baseline might be unstable in real IVT condition.

Response: Thank you for your comments and we are very sorry for missing description of the negative control. In the initial experiment, we examined the effects of different negative controls, such as the omission of DNA templates, NTPs, RNA polymerase, and addition of 5% SDS or 20 mM EDTA respectively. As the results showing in the following Figure A, missing DNA templates or NTPs were the best combinations. Since DNA templates needs to be added first to form a complex with RNA polymerase, we choose omit the NTPs (can be added lastly) as a negative control preferentially. We have added this instruction in Line 213-214, Line 525-529 and Supplementary Fig. S1 of our new manuscript.

In addition, we don't know why EDTA and SDS affect probe fluorescence, the reason for the effect of RNA polymerase was verified: as large molecular weight proteins, RNA polymerase solutions require approximately 50% glycerol to facilitate cryopreservation. The glycerol was simultaneously mixed in an *in vitro* transcription reaction system to affect solution viscosity and thus to affect probe Brownian motion and fluorescence. The effect of glycerol content on probe fluorescence was shown in Figure B below.

2. To deviate quantitative FRET signal, signal of negative control was measure in each assay condition in Fig 3 and 5. If the baseline measurement need to be accompanied, the benefit from multiplex assay is quite damaged. Why does the baseline signal appear changing even without target DNA? Is it possible to set a fixed baseline for final assay condition?

Response: As responded in question one, the omission of NTPs is ideal choice as a negative control for almost no effect on the fluorescence of the probes. However, due to the inevitable existence of testing and manipulation errors, we must test the experimental and negative groups in the same batch. In particular, for multichannel transcriptional activity detection, the experimental groups of all channels and their negative control groups should be tested in the same batch, so a multi-channel RT-PCR thermocycler should be used for the experiment. The results shown that the error value of multiplex assay can be controlled within 6%.

As mentioned, in the early stage of each experiment (about 0 to 8 mins), the negative control also showed a small increase in fluorescence, which was caused by the influence of temperature. When placing in the testing instrument, it takes a few minutes for the room temperature reaction system (about 25°C) to reach the detection temperature of 37 °C, and the increase of temperature increases the light intensity of the fluorescence groups. Preheating instruments and samples in advance can mitigate this effect. However, temperature had the same effect on the experimental group and the negative control group, so the signal corrected results (the negative control minus experimental group) could ignore this interference. Based on the above considerations, we recommend the use of the no NTPs group as a uniform negative control, which we have highlighted in the “Methods” part (Line 525-529) of our new manuscript.

Response to Reviewer #2

Overall impression of the work:

The manuscript by Zhang et al., entitled: "A method for multiplex real-time quantification of in vitro transcriptional mRNA production (RT-IVT)" describes a method that uses binary fluorescence quencher (BFQ) probes and a DNA ligase for the real time monitoring of the produced IVT-mRNA. The method is well-structured, innovative and very interesting. To strengthen the translational potency of this study, I would suggest as follows:

Specific comments:

Comment 1: Please proof-read again and make the appropriate language corrections.

Line 46: realized  conducted

Line 57: please use the abbreviation IVT-mRNA for in vitro transcribed messenger RNA

Line 60: pushing IVT trials  placing IVT-mRNA trials

Line 69: However, these methods all need  However, all these methods need

Line 71: for trace  to trace or for tracing

Line 90: delete an

Line 92: correct realize

Line 94: transcribed mRNA  IVT-mRNA

Line 94: a "binary fluorescence quencher (BFQ)" probes  delete "a"

Line 97: Rephrase "Which..." it is not gram marly correct

Response: Thanks for the positive comments. All above typos have been corrected. Furthermore, the English language of this manuscript has been carefully edited by Dr. Joseph Elliot at the University of Kansas, and grammatical errors were corrected and marked in red in our new manuscript.

Comment 2: The results section must be more compact and this text (Line 109-129) should be relocated at the Methods section and the Discussion section. Perhaps, the authors could consider re-writting the results section.

Response: Thanks for your suggestion. In the new version, we simplified the presentation of the Results section, and integrated the content related to analysis into the Discussion section. In particular, we removed Line 126-138 and Line 334-340, and adjusted Line 367-376 to Line 461-468 of the Discussion section.

Comment 3: Despite the advantage of the real time task, comparison between this method and more conventional approaches like spectrophotometry, or reverse transcription would highlight the importance of this method.

Response: Thanks for your comments. We have compared the detection effect of our RT-IVT method with the spectrophotometry and reverse transcription (RT-PCR) methods as shown in the following figure and in Figure 5e in our manuscript. From the results, the detection results of RT-IVT and spectrophotometry methods are basically the same. As a relative quantitative detection method, the initial slope of the RT-PCR monitoring curve was also similar to the RT-IVT method. We have added the descriptions of the spectrophotometry and reverse transcription methods in Line 325-329 of the new manuscript.

The orange square and red circles in Fig. 5e represent the transcriptional activity results for DNA template 1 tested by the spectrophotometric and the RT-PCR methods. Between the groups, the relative quantitative detection results for the RT-PCR method corresponded to the right, red coordinate axis. The values represent the mean \pm standard deviation of $n = 3$ replicates.

Response to Reviewer #3

In this manuscript, the authors have developed a new method RT-IVT for the real-time measurements of mRNAs in the IVT reaction. Their method leverages the fluorescence quenchers & DNA ligase and uses the FRET and RNA-splinted DNA ligation methods. By using this method, authors have characterized RNA polymerases from bacteria and phages. While the manuscript presents a method improvement; however, I have some reservations. please check my comments below:

Major/minor comments.

1. recommend authors use a uniform font type and size across all figures.

Response: Thanks for your comments, and we have unified the font type and size across of all figures in our new manuscript.

2. Figure legends are too abstract or method oriented. For example Fig2a, the authors have shown a bar graph representing relative flou. ins. values. What do the error bars represent? Have authors performed any statistical tests to check if the decline in flou. values are statistically significant over increasing PBCV-1 concentrations? Please resolve FL-related issues across all figure legends.

Response: In Fig. 2abcde, all of the values shown represented the mean \pm standard deviation of the results from five independent experiments. To avoid duplication, we added this note at the end of the Fig. 2 legends. Similar data replicate descriptions were shown at the end of all other figure legends.

According to your comments, one-way ANOVAs followed by Duncan's multiple comparison were used to test the differences among PBCV-1 concentrations, which were performed at $\alpha = 0.05$. The significant differences are shown in Fig. 2a and 2d in our new manuscript.

3. Panel B, why do the first two rows have two samples with zero PBCV-1 concentrations? if there is any rationale, please mention it in the figure legends.

Response: We are very sorry for the omission of this explanation. In Fig. 2b, the first lane represents BFQ1 added, the second lane represents BFQ1 and cRNA added. We have mentioned it in Fig. 2b in our new manuscript and as follows.

Figure 2. b) The RNA-splinted DNA ligation assays for different concentration of PBCV-1 DNA ligase.

4. Fig 2D, why do authors compute the statistical significance for a few comparisons, not all?

Response: Thanks for your comments. We have computed the statistical significance for all data in Fig. 2d in our new manuscript and shown as follows.

Figure 2. d) The FRET quenching signal of BFQ1a2 with BFQ1b2-BFQ1b6. Different letters indicate significant differences among PBCV-1 DNA ligase concentrations and fluorophores labeling positions adjusted according to Duncan’s test ($p < 0.05$, $n = 5$).

5. I recommend that authors avoid adding methodological details in the figure legends but restrict to the panel explanation.

Response: Thanks for your comments. We have rewritten the figure legends in our new manuscript to avoid adding methodological details. For example, transfer the “RNA-splinted DNA ligation assays” and “probes standard curves detection” explanations to the “Methods” section. However, in order to distinguish the details, we still retain part of the description of the added samples, looking forward to your understanding.

6. Fig 3 and 4, is it one sample? or is it experiment done as a single replicate? if not, please provide the normalized, scaled data of all replicated alongside the statistical test (ANCOVA?).

Response: We are very sorry for the misunderstanding. Fig.4 was intended to test the

tolerance of the RT-IVT method to different NTP, pH and iron conditions, which was not a single replicate of Fig. 3. According to your comments, we analyzed the Fig.4 data statistically (one-way ANOVAs followed by Duncan's multiple comparison) and adjusted the corresponding figure legends in our new manuscript and shown as follows.

Figure 4. Effects of reaction buffers on RT-IVT assays. The relative initial transcription velocity of T7 RNA polymerase at different NTP (a), pH (b), NaCl (c), KCl (d), MgCl₂ (e) and MnCl₂ (f) concentrations. Except for the NTP optimization experiment, the NTP concentration was 250 μM for the other groups. The shaded areas represent the common concentration range for each ion, and triangles represent the ion concentrations used. The values shown were the mean \pm standard deviation of three repeats. Different letters indicate significant differences among NTP, pH, or ion treatments according to Duncan's test ($p < 0.05$, $n = 3$).

7. Authors have made interesting comments in the abstract/manuscript, i.e., their method is cheaper than other methods. I recommend authors revisit this claim and do provide substantial evidence (suppl. info) for this.

Response: According to your suggestion, we have compared the cost of the current popular IVT-mRNA detection methods in Line 331-334 and Table 2 in our new manuscript and as follows. Among them, the synthetic cost of the BFQ probes was 600 RMB per 1000 reactions. The cost of PBCV-1 DNA ligase purchased from NEB company was 1250 RMB per 50 reactions, and the cost of it purified in our laboratory was 100 RMB per 1,000 reactions. After rate calculation of RMB to USD, the detection cost of RT-IVT method were 3.804 or 0.104 \$ per reaction. According to the above results, we retain the statement of price advantage but replaced “cheaper” with “inexpensive” in the manuscript, looking forward to your understanding.

Table 2. The collections of the current popular IVT-mRNA detection methods.

Detection methods	Duration	Price/reaction ^a	Detection components	Instrument	Origins
RT-IVT	3 hours	3.804 or	BFQ probes (Sangon Biotech)	RT-PCR thermocycler	This study
	(real-time)	0.104 ^b	PBCV-1 DNA ligase (NEB)	or microplate reader	
Spectrophotometry	5 hours	4.458	RNA purification kit (TaKaRa)	Spectrophotometer	(11)
RT-PCR	8 hours	14.117	RNA purification kit (TaKaRa)	RT-PCR thermocycler	(10)
			cDNA Synthesis kit (TaKaRa)		
			SYBR Premix PCR kit (TaKaRa)		
Radioactively tagged nucleotides replacement	12 hours	14.860	[α - ³² P] UTP mixed with ATP, CTP, and GTP (PerkinElmer) Denaturing gel electrophoresis	Radiological safety laboratory and scintillation counter	(12)

^aThe cost only applies to the detection components in the reaction and the price was transferred by rate calculation of RMB to USD.

^bThe price of 3.804 was calculated using BFQ probes added to PBCV-1 DNA ligase purchased from New England Biolabs (NEB) corporation, while the price of 0.104 was calculated using BFQ probes added to PBCV-1 DNA ligase purified in our laboratory.

REVIEWERS' COMMENTS:

Reviewer #1 (Remarks to the Author):

the raised issues on negative control condition were well-explained with additional data and description.

Reviewer #3 (Remarks to the Author):

The authors have addressed all my concerns. I congratulate them for their wonderful work.